# Detection and Genome Sequencing of Lumpy Skin Disease Viruses in Wildlife Game Species in South Africa

**DOI:** 10.3390/v16020172

**Published:** 2024-01-24

**Authors:** Antoinette van Schalkwyk, Pravesh Kara, Robert D. Last, Marco Romito, David B. Wallace

**Affiliations:** 1Agricultural Research Council—Onderstepoort Veterinary Institute, Pretoria 0110, South Africa; karap@arc.agric.za (P.K.); romitom@arc.agric.za (M.R.);; 2Department of Biotechnology, University of the Western Cape, Bellville 7535, South Africa; 3Department of Biochemistry, Microbiology & Genetics, University of Pretoria, Pretoria 0110, South Africa; 4Vetdiagnostix–Veterinary Pathology Services, Pietermaritzburg 3200, South Africa; rick@vetdx.co.za; 5Department of Veterinary Tropical Diseases, Faculty of Veterinary Science, University of Pretoria, P/Bag X4, Pretoria 0110, South Africa

**Keywords:** lumpy skin disease virus, springbok, giraffe, complete genome sequencing, phylogenetics, single nucleotide polymorphisms (SNPs)

## Abstract

Lumpy skin disease virus (LSDV) has recently undergone rapid spread, now being reported from more than 80 countries, affecting predominantly cattle and to a lesser extent, water buffalo. This poxvirus was previously considered to be highly host-range restricted. However, there is an increasing number of published reports on the detection of the virus from different game animal species. The virus has not only been shown to infect a wide range of game species under experimental conditions, but has also been naturally detected in oryx, giraffe, camels and gazelle. In addition, clinical lumpy skin disease has previously been described in springbok (*Antidorcas marsupialis*), an African antelope species, in South Africa. This report describes the characterization of lumpy skin disease virus belonging to cluster 1.2, from field samples from springbok, impala (*Aepyceros melampus*) and a giraffe (*Giraffa camelopardalis*) in South Africa using PCR, Sanger and whole genome sequencing. Most of these samples were submitted from wild animals in nature reserves or game parks, indicating that the disease is not restricted to captive-bred animals on game farms or zoological gardens. The potential role of wildlife species in the transmission and maintenance of LSDV is further discussed and requires continuing investigation, as the virus and disease may pose a serious threat to endangered species.

## 1. Introduction

Lumpy skin disease (LSD) is a relatively recent disease of cattle, since the first symptoms were described in Northern Rhodesia (now, Zambia) in 1929 [1]. Neither large-scale commercial nor local indigenous farmers had ever reported the clinical signs associated with the disease in their herds before. Initially, it was assumed that these signs were linked to insect bites, plant toxins or the use of newly introduced pesticides [2]. However, the rapid spread of the disease throughout most of Africa over the ensuing years, led to its confirmation and characterisation as being caused by an infectious agent, lumpy skin disease virus (LSDV), a poxvirus [3]. Since 1989, the disease has been confirmed in Israel and the rest of the Middle East, and subsequently in Turkey, the Balkans, Eastern Europe, Russia, Kazakhstan, and more recently it has spread to large parts of eastern and southern Asia [4,5,6,7]. 

LSDV belongs to the *Capripoxvirus* genus in the *Poxviridae* family, along with goatpox (GTPV) and sheeppox viruses (SPPV). All three viruses share a common major precipitating antigen [8] making it impossible to distinguish between them based on currently available serological techniques [9]. Similarly, the three capripoxviruses share more than 93% sequence identity across their entire genomes [10]. Prevention and control of LSD in cattle is based on the implementation of vaccination programmes using either live attenuated vaccines (LAV), which are homologous (“Neethling-type” LSD vaccines), or heterologous vaccines utilising sheeppox or goatpox virus vaccine strains [11].

Southern Africa is free from GTPV and SPPV, and LSDV has only been isolated from cattle in the sub-region [12,13], whilst in the rest of Africa it is not uncommon for all three capripoxviruses to occur in the same geographic region, with cross-species infections reported between cattle, sheep and goats [14,15]. In addition to cattle, domesticated water buffalo (*Bubalus bubalis*) in Egypt have been reported to be mildly susceptible to natural infection with LSDV [16,17]. In contrast, giraffes and impala showed a high degree of susceptibility when experimentally infected, while African buffalo and black wildebeest (*Connochaetes gnu*) showed no adverse effects [18].

In 1989, the first detection of a capripoxvirus in a naturally infected game animal, the Arabian oryx (*Oryx leucoryx*), was described [19]. This was in conjunction with capripoxvirus antibodies detected in the same group of captive animals during an earlier survey [20]. However, no further genetic characterisation was performed to determine the species of virus involved. LSDV DNA was detected in springbok in South Africa in the early 2000s [21] and in an eland antelope sampled in 2019 (*Taurotragus oryx*) in Namibia, which was asymptomatic for LSD [22]. In 2022, the isolation and characterisation of LSDV from a giraffe in a zoo was reported in Vietnam [23]. With the spread of LSD to Asia, new local wildlife species have also been found to be susceptible to the disease. These included gaurs (*Bos gaurus*), mainland serow (*Capricornis sumtraensis*) and banteng (*Bos javanicus*) in Thailand and Cambodia in 2021 [24]. Complete genome sequencing of LSDV was performed directly from clinical samples submitted from yaks in China, where the LSDV caused high mortality within the yak population [25]. Additionally, LSDV has been isolated and genetically characterised in clinically affected camels and two free-ranging Indian gazelles (*Gazella bennettii*) in India, in 2022 [26,27]. 

Serological surveys of game animals in South Africa, Kenya and Tanzania detected antibodies to capripoxvirus in African buffalo (*Syncerus caffer)*, greater kudu (*Tragelaphus strepsiceros*), waterbuck (*Kobus ellipsiprymnus* and *K. defassa*), reedbuck (*Redunca arundinum*), impala (*Aepyceros marsupialis*) wildebeest (both species: blue, *Chonnochaetes taurinus* and black, *C. gnou*), springbok, eland and giraffes [19,28,29,30,31]. A recent seroprevalence study in African buffalo in two wildlife reserves in the north-east of South Africa, indicated that 7 to 28% of buffalos have low levels of antibodies to LSDV, depending on the type of serological test used [32]. This raised the possibility that buffalos and other wildlife species may serve a role as natural and/or long-term maintenance hosts [32]. Further evidence in support of this was found in a survey in Zimbabwe on the distribution of LSD, where higher incidences of LSD outbreaks were recorded in cases where cattle were close to game species [33].

On rare occasions, reports in South Africa of LSD-like lesions in various game animal species, including kudu, giraffe, African buffalo and gemsbok (South African oryx, *Oryx gazella*), have emerged. However, attempts to isolate and/or identify LSDV from these animals were unsuccessful (Wiese, personal communication, 2008). However, the scenario in springbok is different. Clinical signs typical of LSD, including cutaneous nodules, lymphadenopathy and pyrexia, are quite frequently encountered, especially in captive-bred animals. The cutaneous lesions are generally firm, circumscribed nodules measuring approximately 0.5–5.0 cm in diameter with lesion distribution similar to that described in cattle—namely the head, neck, limbs, udder, genitalia and perineum [34]. Nodules affecting the scrotum, perineum, udder, vulva, glans penis, eyelids, and conjunctiva are usually flatter. Typically, nodules undergo necrosis and sequestration, but some may resolve rapidly and completely, while a few may become indurated and persist as hard intradermal lumps for many months. There is a high risk of secondary bacterial infection during the period of necrosis and sequestration. Detection of LSDV in a number of South African springbok samples has been described using PCR and gene-specific sequence analysis [21,35], although no complete genome sequencing was performed on these samples.

This study describes the molecular detection, characterization and complete genome sequencing of LSDV obtained from springbok and giraffe samples submitted from national game parks and private game reserves in South Africa. Lesion or blood samples were sent to the Agricultural Research Council, Onderstepoort Veterinary Institute (ARC-OVI) for virus identification and subsequent analysis. In addition to using PCR to amplify viral DNA, Sanger sequencing of partial open reading frames (ORFs) and subsequent complete genome sequencing were performed directly from virus DNA (vDNA) extracted from lesions and/or blood [36,37]. The partial ORFs and complete genome sequences were subjected to phylogenetic analysis to establish the relationship of these LSDV strains to other capripoxviruses.

## 2. Materials and Methods

### 2.1. Sample History 

In addition to cattle samples, samples from various clinically affected wildlife species were submitted to the ARC-OVI for laboratory confirmation of LSDV between 2000 and 2017. These samples were submitted from four of the nine South African provinces (Table 1).

In May 2000 and June 2006 two batches of samples were sent to the ARC-OVI Laboratory for diagnosis, taken from springbok on farms near Kimberley in the Northern Cape (NC) Province of South Africa (Table 1). In the 2000 case (sample: LSDV_SB01-NC_RSA_2000), a skin biopsy sample was taken from a live animal. Additionally, seven other springboks in the same herd displayed similar LSD-like lesions, but no samples were collected from them. These animals were in close proximity to cattle, which were not vaccinated for LSD, but did not display LSD-like symptoms. In the 2006 case (sample: LSDV_SB02-NC_RSA_2006), lesions were observed on the scrotum and other areas of the body of a springbok that was sent for slaughter. Lesions from the lungs, skin, lymph nodes and testes were removed and submitted for diagnosis to the ARC-OVI.

In 2011, samples from two individual springbok antelopes were submitted from a game farm in the Rustfontein dam area and the Gariep Nature Reserve (National Park) in the Free State Province. Both samples were from adult male animals. Sample SB11-FS_RSA_2011 only contained skin nodules, whilst sample SB2366-FS_RSA_2011 consisted of skin and internal organs (lung, kidney, heart and lymph node) (Table 1, Figure 1). In August 2011, a tissue sample from an impala antelope was submitted from a game reserve in Mpumalanga, which borders the Kruger National Park (LSDV_IP4318-MP_RSA_2011).

Skin nodules from a 3-year-old female giraffe (LSDV_G12-KZN_RSA_2012) in the Hluhluwe Nature Reserve in the KwaZulu-Natal province were submitted in April 2012 (Table 1). 

Finally, in 2017, skin lesions from a springbok displaying LSD-like lesions were submitted from a game farm near Parys in the Free State Province (sample LSDV_SB184-FS_RSA_2017). In addition, EDTA blood samples were sent from two different game farms in the North West Province in September 2017 (LSDV_SB259-NW_RSA_2017 and LSDV_SB260-NW_RSA_2017) [34] (Table 1). 

### 2.2. Laboratory Confirmation of LSDV and Sanger Sequencing

Thin-tissue sections were prepared from each of the lesion samples, ground using a mortar and pestle in phosphate-buffered saline and clarified using low-speed centrifugation. The clarified supernatants were used for DNA extraction and PCR amplification.

DNA was extracted from the samples using the MagNA Pure 96 (Roche, Molecular Systems Inc., Heidelberg, Germany) automated robotics extractor system according to the manufacturer’s instructions. The presence of LSDV genomic DNA was detected using the previously described method based on the viral thymidine kinase (TK) gene [36].

Samples positive for LSDV DNA were subsequently characterized using additional PCR amplification and Sanger sequencing of the partial ORF of LW036, as previously described [37]. In short, 2 μL of the previously extracted DNA was used in a 20 μL reaction containing 10 μL of 2× Dreamtaq DNA polymerase master mix (Life Technology, Carlsbad, CA, USA), with 0.25 µmol/L of each primer (LW036-F: TAT GTT ATT TTT CTA CAG CTC TAA and LW036-R: CAG TAC AAA CAT GGA TGA TGA T) at an annealing temperature of 53 °C for 45 cycles. An aliquot of the resulting amplicons was analysed using 1% agarose gel electrophoresis and the remainder of the reactions were submitted to Inqaba Biotechnical Industries (Pretoria, South Africa) for Sanger sequencing, using the primers incorporated during the generation of the amplicons. Sequence data were analysed and compared using CLC Genomics Workbench 9.5 (QIAGEN Aarhus, www.clcbio.com (accessed on 30 November 2016)) and a Maximum Likelihood phylogenetic tree was generated using General Time Reversal (GTR) (G + I, G = 4) with 1000 bootstrap iterations in Mega X, as previously described [37,38]. 

### 2.3. Complete Genome Sequencing 

Virus DNA was extracted directly from skin lesions using the protocol previously described with the following modifications [39]. Tissue sample (50 × 50 × 50 mm) was dissected from the skin nodules and transferred to a hard tissue homogenising tube (Precellys, Bertin Technologies, Montigny-le-Bretonneux, France). The tissue sample was covered with 0.9 mL hypotonic buffer (10 mM Tris, pH 8.0, 10 mM KCl, 5 mM EDTA) and placed in a TissueLyser LT (Qiagen, Hilden, Germany) for 8 min at 50 oscillations per second. The cellular debris was removed by centrifugation at 2000 rpm for 5 min at 4 °C. The supernatant was transferred to a clean Eppendorf tube and incubated at 0 °C for 10 min, followed by the addition of 25 μL 2-mercaptoethanol and 10% (*v*/*v*) Triton X-100. The sample was left overnight at 0 °C with gentle mixing followed by centrifugation at 13,000 rpm for 60 min at 4 °C. The pellet was resuspended in 80 μL cold buffer (10 mM Tris pH 8.0, 1 mM EDTA) mixed with 15 μL of 2-mercaptoethanol, 5 μL proteinase-K (500 pg) and 20 μL 20% (*w*/*v*) N-lauroyl sarcosinate (NLS, Sigma-Aldrich, Burlington, MA, USA), and incubated at 4 °C for 30 min. Thereafter, 14 μL of 54% (*w*/*v*) sucrose and 40 μL 5 M NaCl was added to the sample and the mixture was incubated overnight at 55 °C. DNA was purified through three rounds of extraction with an equal volume of phenol:chloroform:isoamyl alcohol (50:48:2). The purified DNA was subsequently submitted to the Agricultural Research Council—Biotechnology Platform (ARC-BTP) for NGS. The concentration and quality of vDNA was examined on a 2100 Bioanalyzer (Agilent, Santa Clara, CA, USA) and only samples (*n* = 3) with >200 ng vDNA were selected for NGS. A total of 200 ng of purified DNA was fragmented between 100 and 300 bp fragment sizes using the Covaris ME220 focused ultrasonicator (Covaris, Woburn, MA, USA) according to the manufacturer’s instructions. For each final prep, 25 ng of DNA was processed according to the MGIEasy Universal DNA Library Prep Set (MGI Tech Co. Ltd., Shenzhen, China) protocol and the pair-end 150 sequencing protocol generated datasets comprising ~10 million paired reads per sample using the DNBSEQ-G400 platform (MGI Tech Co., Ltd., Shenzhen, China).

### 2.4. Bioinformatics and Phylogenetic Analysis

The reads were mapped against the LSDV_Warmbaths_RSA_2000 sequence (GenBank accession number: AF409137) using the CLC Genomics Workbench v9 (Qiagen, Hilden, Germany) software package. The average coverage of the mapped reads ranged from 30 to 500 across the reference genome sequence, resulting in the generation of a single consensus sequence per sample. These newly generated consensus sequences were deposited in GenBank under the following accession numbers: OR644282 to OR644284.

Alignments were generated using other capripoxvirus sequences available on GenBank in CLC Genomics Workbench v9 (www.clcgenomics.com (accessed on 30 November 2016)). The phylogenetic relatedness of capripoxviruses was investigated using the complete genome sequences. Maximum likelihood phylogenetic trees were constructed under General Time Reversal (GTR) (G + I, G = 4) with 1000 bootstrap iterations in Mega X as previously described [37,38]. 

## 3. Results

### PCR, Sanger and Complete Genome Sequencing

Since the year 2000, bovine (*n* = 483), ovine (*n* = 40), caprine (*n* = 2), African buffalo (*n* = 4), Dorcas gazelle (*n* = 17), horned Oryx (*n* = 10), zebra (*n* = 2), wildebeest (*n* = 2), sable (*n* = 2), kudu (*n* = 2), giraffe (*n* = 13), springbok (*n* = 15), and impala (*n* = 2) samples have been submitted to the ARC-OVI for molecular detection of vDNA using LSDV-specific primers [36]. Although the majority of the wildlife samples were intended for export and therefore expected to be PCR-negative, 199 bovine, as well as springbok (*n* = 7), giraffe (*n* = 1) and impala (*n* = 1) samples tested positive for LSDV. 

The LSDV-positive springbok, impala and giraffe samples came from animals that displayed clinical signs (Table 1). In the year 2000, the South African Department of Agriculture reported a total of 588 LSD outbreaks, with 24 of them detected in the Northern Cape Province [40]. The month with the highest incidence of LSD outbreaks was May 2000 (*n* = 24), with outbreak reports in cattle, accompanied with samples, received from Britstown, Carnavon, Colesberg, De Aar, Philipstown and Prieska, whilst the springbok sample (LSD_SB01-NC_RSA_2000) was collected on a game farm in the Kimberly district (Table 1). Additionally, only two LSD outbreaks were reported in June 2000, both in cattle from the Kimberly district. In contrast to the large number of LSD outbreaks in the Northern Cape Province in 2000, only seven LSD outbreaks were reported in the province in 2006. The majority (*n* = 3) were in March, whilst a single outbreak in bovines was reported in May and one from springbok (LSDV_SB02-NC_RSA_2006) in June, both from the Kimberley district (Table 1) [40]. Unfortunately, no genetic information pertaining to either the springbok or cattle samples for these outbreaks are available (Table 1). 

The South African Department of Agriculture reported 213 outbreaks in 2011, the majority (*n* = 58) occurring in March and (*n* = 57) in April [40]. Skin nodules from two individual springbok antelopes (SB11-FS_RSA_2011 and SB2366-FS_RSA_2011) submitted from the Free State Province were used to extract vDNA using the modified protocol described within this study (Table 1, Figure 1). Sufficient quantity and quality of vDNA were obtained for library preparation and NGS. Unfortunately, vDNA extracted from a tissue sample from an impala antelope (LSDV_IP4318-MP_RSA_2011) submitted in August 2011 from Mpumalanga was not sufficient for NGS and subsequently only Sanger sequencing of the partial LW036 ORF was attained. The outbreaks in the Free State were associated with both cattle and springbok **(**Figure 2). The cattle samples in the Free State province were confirmed as LSDV-positive using the same TK-specific PCR method (no additional molecular characterization of these samples was performed), but Sanger sequences of the partial LW036 ORF were obtained from bovine samples from the North West and Mpumalanga Provinces in 2010 (Figure 3). Additionally, of the 166 LSD outbreaks reported in 2017, three were from springbok in the Free State and North West Province (Table 1). A sequence of the partial LW036 ORF was generated from the sample (LSDV_SB260_NW_RSA_2017) and compared to the same gene region of other capripoxviruses. Based on the partial LW036 ORF, the bovine, springbok and impala samples were related with 100% sequence identity over the 606 bp region, and the sequences all grouped together in Cluster 1.2 (Figure 3).

As was the case for the two springbok samples from 2011, vDNA extracted from a giraffe sample in 2012 (G12-KZN_RSA_2012) produced sufficient vDNA for NGS (Table 1). Complete ~150,200 bp genomes were generated from these three samples: SB11-FS_RSA_2011, SB2366-FS_RSA_2011 and G12-KZN_RSA_2012. The complete genome sequences were used to investigate the phylogenetic relationship of these LSDVs with other strains of capripoxvirus (Figure 4). The wildlife game sequences grouped with other LSDV field strains in cluster 1.2 (Figure 4). Both springbok and giraffe sequences shared between 99.95 and 100% sequence homology with all other recent LSDV field strains from South Africa and are distinct from the Neethling vaccine, recombinant LSDVs, SPPV and GTPV strains. 

## 4. Discussion

LSD was initially and most frequently encountered in cattle, and thus the virus was thought to be primarily host-range restricted to them [12]. As the virus spread throughout Africa, it was additionally observed to be a milder disease in domesticated water buffalo, but since they do not occur widely in southern Africa, they were thought unlikely to be the natural host. Despite a large sheep and goat population in this subregion, the disease has never been described in local sheep or goats [13]. Since the disease was first described in Zambia and remained confined to sub-Saharan Africa for decades, it is likely that the natural host species is native to the subcontinent. The Bovidae family, which includes cattle, yaks, sheep, goats and antelopes, is estimated to have evolved 20 million years ago. There are approximately 143 extant and 300 known extinct species within this family, with the African continent having the largest diversity of antelope species [41]. It has been calculated that LSDV diverged from the other capripoxviruses around 12,000 years ago and the two main clusters (1.1 and 1.2) diverged around 500 years ago [42,43]. This suggests that LSDV theoretically diverged prior to the predicted domestication of cattle, which occurred about 10,000 years ago, and before the introduction of cattle to Africa which happened about 5000 ago [44]. Therefore, it is highly plausible that LSDV evolved in one or more antelope species within the *Bovidae* family, rather than in cattle.

Sporadic reports of LSD-like lesions in wildlife ruminant species as well as the occasional clinical samples have been received since its first reporting in cattle. However, direct evidence for the presence of the virus in these species was generally lacking and the potential significance of the disease occurring in them went largely unexplored. Experimental infection of impala, giraffe, African buffalo and black wildebeest [18] indicated a high-level of susceptibility in impala and giraffe, but not so for buffalo or wildebeest. However, isolation and characterisation of the virus from naturally infected game species has proved to be more problematic. This is often due to difficulties in obtaining samples from wild animals in large game parks or nature reserves, which often fall within FMD-restriction zones. Since the majority of the samples received from wildlife species were during the capture of healthy animals for export purposes, the actual extent of the disease prevalence in these animals is unknown. However, the recent introduction of captive-breeding of many game species on farms or in zoos in the absence of predators, has made it more frequent and practical to observe and sample species displaying clinical disease. Greth et al. (1992) reported on capripox infection in captive-bred Arabian oryx [19] and Dao et al. (2022) described the characterisation of LSDV from a giraffe in a zoo in Vietnam [23], amongst others. The recent and rapid spread of LSD into Asia has resulted in an increasing number of reports of other ruminant species showing susceptibility to infection with the virus [25,26,27]. This is a major concern as a number of the species involved are endangered. 

Although lesions typical of LSD have been reported in species such as kudu, African buffalo and gemsbok, identification of the virus from skin lesions from these animals has been relatively rare. However, this is not the case for springbok, and more recently, giraffe. It is possible that the occurrence of LSD in springbok, under relatively natural conditions and with a limited number of natural predators, is more frequent than previously thought. In 2019, during a survey of vector-borne diseases and wild game species in Botswana in the Central Kalahari Game Reserve, using camera traps several instances of springbok displaying LSD-like lesions were captured in different individuals (Christe, Buxton and Nyamukondiwa, unpublished) (Figure 1A). Springbok and impala are common throughout southern Africa and reproduce quickly, while maintaining giraffes successfully requires more restrictive conditions. 

Limited characterisation of LSDV from natural infections of springbok in South Africa has been ongoing [21,34,35], but only recently has full characterisation at the genetic level from springbok antelope, as described in this paper, been practical. Additionally, we reported here on viral genomic DNA amplification and full-genome sequencing of LSDV from a lesion sample from a in the Hluhluwe Game Reserve in KwaZulu Natal. This is mainly due to advances in sequencing technology enabling full-genome sequencing using low amounts of genomic DNA as starting material, without the need to first isolate and propagate the virus.

The LSDVs from springbok and giraffe that were fully sequenced and compared to other isolates of LSDV in this paper displayed 99–100% sequence homology to cluster 1.2 strains, which includes the Warmbaths/2000 isolate. Additionally, the sequences of the partial LW036 ORF from cattle, springbok and impala confirmed that the dominant circulating strain of LSDV belongs to cluster 1.2 (Figure 3). Cluster 1.2 strains were first detected in southern Africa after 2000, prior to which only cluster 1.1 isolates were detected (prototype, Neethling and including the Neethling Onderstepoort vaccine strain) [45]. No solid evidence is yet available to help explain this relatively rapid, and apparently complete replacement of cluster 1.1 with cluster 1.2 genomes. Phylogenetic mapping of the LSDV field and vaccine genomes may provide at least part of the answer (Figure 4). Since cluster 1.2 genomes have been isolated from LSDV strains in Kenya in the 1950s, it is possible that an animal harbouring the virus and asymptomatic for LSD was introduced into southern Africa from East Africa in the late 1990s. 

LSD viruses have been detected and characterised in giraffes, springbok and impala in South Africa. These viruses were identical to the strains currently circulating in cattle. The role that these species play in the long-term maintenance of the virus remains to be determined, but as the virus continues to spread and is reported in more animal species, concerns are being raised as to the damaging impacts these outbreaks may have on native, and especially, endangered species. 

Vaccination remains the most effective and practical method for controlling and preventing the disease. However, implementing vaccination programmes in wildlife species in their natural environment will be challenging. Work on safer and improved vaccines is continuing and with concerted efforts between all role players in the wildlife and domestic livestock fields the relevant solutions to reducing the impacts of the disease will most likely be found.

## Figures and Tables

**Figure 1 viruses-16-00172-f001:**
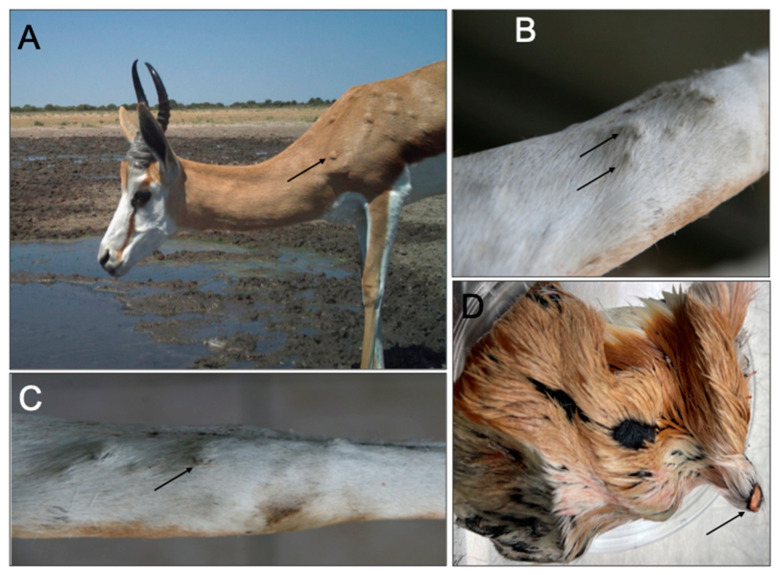
Clinical signs of LSD on various springbok animals. (**A**) Picture of a springbok with LSD-like nodules (example of a nodule indicated with an arrow) from a camera-trap submitted by P. Christe. (**B**,**C**) Skin nodules (arrows) on the legs of springbok sample SB2366-FS_RSA_2011. (**D**) Skin lesion (arrow) of sample SB11-FS_RSA_2011 used during the extraction of vDNA for NGS.

**Figure 2 viruses-16-00172-f002:**
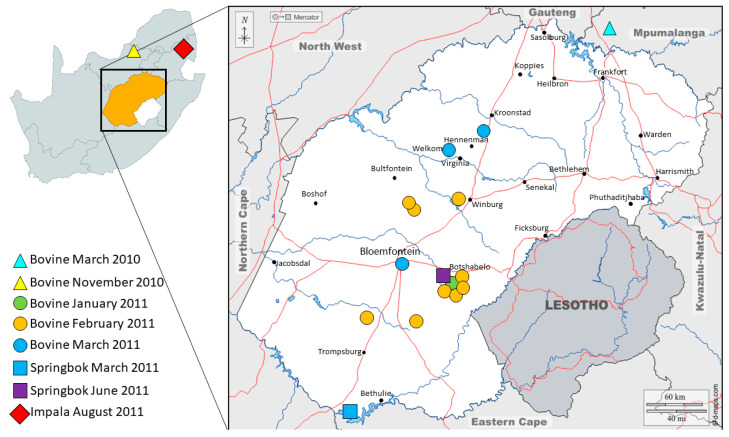
Map of South Africa, with the Free State province (in orange) highlighted in the square, indicating the LSDV PCR-positive cases from cattle and springbok. The cattle samples are indicated with circles if they were only PCR positive, whilst the samples that were partially sequenced are indicated with triangles. Complete genome sequences of the LSDVs were performed for the two springbok samples indicated with squares. The location of the impala sample is indicated with a red diamond.

**Figure 3 viruses-16-00172-f003:**
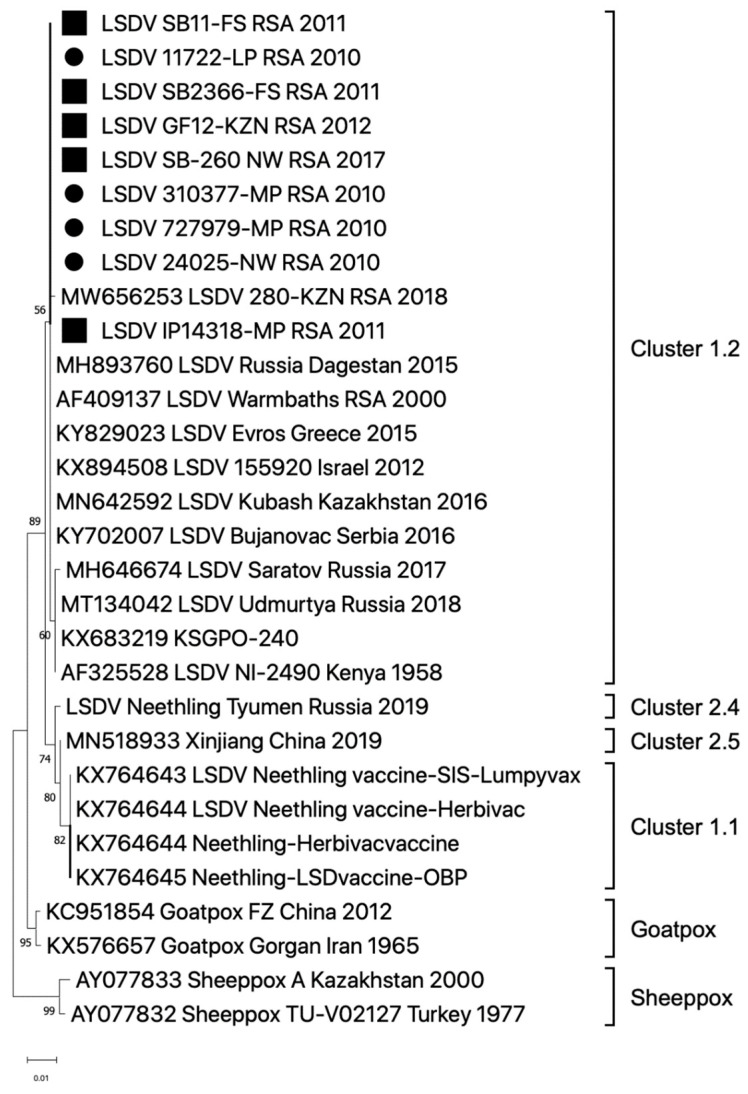
Phylogenetic relatedness of LSDVs based on the partial ORF LW036 sequences, as compared to sheeppox and goatpox viruses. The sequences of three springbok, one giraffe and one impala sample are indicated with black squares, whilst the bovine samples are indicated with circles. The maximum likelihood phylogenetic tree was constructed in Mega X.

**Figure 4 viruses-16-00172-f004:**
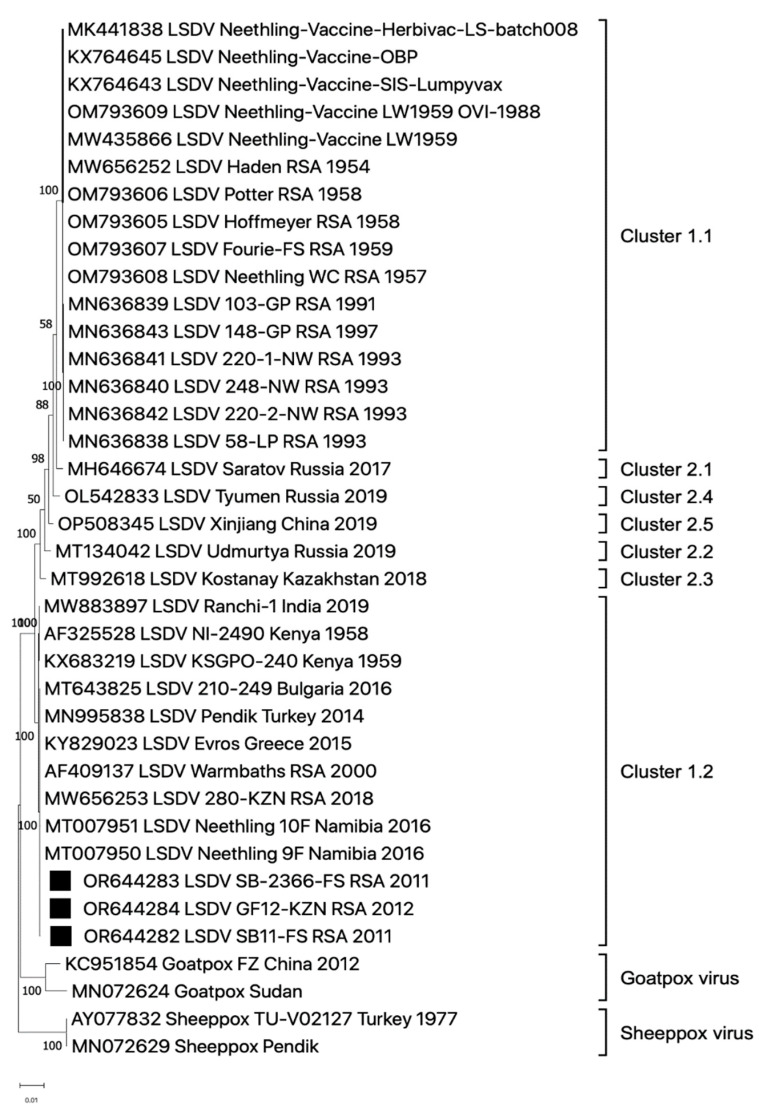
Phylogenetic relatedness of LSDVs based on their complete genome sequences, as compared to sheeppox and goatpox viruses. The complete genomes of two springbok and one giraffe sample are indicated with black squares. The maximum likelihood phylogenetic tree was constructed in Mega X.

**Table 1 viruses-16-00172-t001:** Sample information and test results obtained for LSDV isolation and characterisation from game species.

Sample Name	Animal Species (Common Name) and Sample Type	Date (Month and Year)	Location (Province of RSA)	LSDV PCR Positive	Complete Genome Sequences (GenBank Accession Number)
LSDV_SB01-NC_RSA_2000	Springbok Skin nodules	May 2000	NC (Game farm close to Kimberley)	Yes	No
LSDV_SB02-NC_RSA_2006	Springbok (adult male) Lesions from skin, lung, testes and lymph nodes	June 2006	NC (Game farm close to Kimberley)	Yes	No
LSDV_SB11-FS_RSA_2011	Springbok (adult male) Skin nodules	March 2011	FS (Gariep dam Nature Reserve)	Yes	Yes (OR644282)
LSDV_SB2366-FS_RSA_2011	Springbok Skin nodules	June 2011	FS (Game farm close to Rustfonteindam)	Yes	Yes (OR644283)
LSDV_IP14318-MP_RSA_2011	Impala Skin nodules	August 2011	MP (Game reserve close to Kruger National Park)	Yes	LW036 only
LSDV_G12-KZN_RSA_2012	Giraffe (3-year-old female) Skin nodules	April 2012	KZN (Hluhluwe game reserve)	Yes	Yes (OR644284)
LSDV_SB184-FS_RSA_2017	Springbok Tissue samples	February 2017	FS (Game farm close to Parys	Yes	No
LSDV_SB259_NW_RSA_2017	Springbok (adult female) EDTA blood	September 2017	NW (Game farm close to Brits)	Yes	No
LSDV_SB260_NW_RSA_2017	Springbok (sub-adult female) EDTA blood	September 2017	NW (Game farm close to Rustenburg)	Yes	LW036 only

SB—springbok, GF—giraffe, IP—impala, NC—Northern Cape, FS—Free State, KZN—KwaZulu-Natal, NW—North West Province, RSA—Republic of South Africa.

## Data Availability

Sequences of assembled and annotated genomes are available at GenBank under the accessions: OR644282 to OR644284.

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
