# Peer review of "Detection and Genome Sequencing of Lumpy Skin Disease Viruses in Wildlife Game Species in South Africa"

_viruses, 2024, doi:10.3390/v16020172_

Round 1

Reviewer 1 Report

Comments and Suggestions for Authors

In the study, it was shown that the lumpy skin disease virus can infect springbok, impala and a giraffe. Whole genome sequences of LSDV were obtained from two springboks and one giraffe  which is undoubtedly very valuable. The manuscript is rather well written and presented in well-structured manner and is interesting. The tables and figures are rather clear presented. My main comment is to improve materials and methods section. Conclusions also should be revised.

Specific comments:

1.      Describing the sample history is not the result. This description should rather be in the "materials and methods" section.

2.      section “sample history” The result described and shown in Figure 3 should be removed from this section.

3.      The results section lacks information on how many samples were tested (9?), how many samples were positive and how many were negative (by each method).

4.      On what basis were the samples selected for NGS?

5.      Rather, the summary of the discussion should be focused on the research results obtained in the work and the conclusions. This needs correction.

Author Response

Thank you kindly for reviewing the manuscript and providing suggestions to improve the manuscript. The comments have been addressed as follows:

Specific comments:

  1. Describing the sample history is not the result. This description should rather be in the "materials and methods" section.

Answer: The section has been moved to the Materials and methods as suggested

  1. section “sample history” The result described and shown in Figure 3 should be removed from this section.

Answer: The section has been moved to the Materials and methods as suggested

  1. The results section lacks information on how many samples were tested (9?), how many samples were positive and how many were negative (by each method).

Information on the number of samples tested is included in the Results section

  1. On what basis were the samples selected for NGS?

Answer: The criteria for sample selection (sufficient quality and quantity: >200 ng vDNA) has been included.

  1. Rather, the summary of the discussion should be focused on the research results obtained in the work and the conclusions. This needs correction.

Answer: The discussion was changed to focus on the work and the results obtained.

Reviewer 2 Report

Comments and Suggestions for Authors

Congratulations and thank you for conducting the present study.

The current research article reports the Lumpy skin disease virus (LSDV) detection and characterization from field springbok, impala, and giraffe samples. This study highlights the presence of LSDV in wildlife of Africa. Full genome sequencing of the LSDV will serve as basis for future studies. 

Review of existing literature is properly done as well as gaps in the current knowledge is identified. Selected research project is relevant to identified gap in current topic. References cited are adequate and represent latest literature. 

Some of the concerns are as follows:

1.) Abstract can be improved by adding the one line about the "Cluster 1.2 genomes in all the three full-length LSDV."  

2.) Was cattle sample(s) part of the current research? While reading the manuscript it is creating little confusion (Line 194-195). Please mention it in the materials and methods section.

3.) What are the chances of introducing DNA mutation while using Dreamtaq DNA polymerase as opposed to high fidality DNA polymerase? Is it of any major concern in the present study?

4.) As current data is strongly suggestive of LSDV is being transmitted between cattle and wildlife. Was LSDV detection attempted in the cattle/wildlife/vectors?

5.) A separate "conclusions" heading may help the readers. Conclusions may be limited to the current experimental findings.  

Specific comments:

Line 4, 6, 7: full stops can be deleted.

Line 123, 124: use "μL" instead of "ul". Use a single space between a number and the unit.

Line 127: use a single space between a number and the unit.

Line 138, 139: use 10 mM instead of 10mM, pH 8 instead of pH8, 10 mM instead of 10mM, 5 mM instead of 5mM. 

Line 141, 142, 143, 144: Keep a space between a number and °C.

Line 142: use 25 μL instead of 25μl.

Line 145-148: use space between a number and unit μL. Use μL instead of μl.

Line 145: use 10 mM instead of 10mM, pH 8.0 instead of pH8.0, 1 mM instead of 1mM. 

Line 146: use 500 pg instead of 500pg.

Line 147: use 4 °C instead of 4°C.

Line 148: use 5 M instead of 5M. 

Line 152: please define "vDNA."

Line 153: use 200 ng instead of 200ng, 300 bp instead of 300bp.

Line 155: use 25 ng instead of 25ng.

Line 174: "is presented in" can be deleted.

Table 1: Date column: Aug can be used to create uniformity instead of August.

Table 1: Animal species and sample type column: what was the sample type for raw LSDV_SB2366-FS_RSA_2011? Suggesting a colon between animal species sample type. Can "common animal name" be used as column heading instead of animal species?

Table 1: Full form of NW is missing in the table abbreviations.

Following sections between text and references need to be added to provide more transparency and clarity for the audience. 

Author Contributions

Institutional Review Board Statement

Informed Consent Statement

Comments on the Quality of English Language

Overall quality of the English language is satisfactory except some minor errors (previously mentioned). 

Author Response

Thank you kindly for reviewing the manuscript and providing suggestions to improve the manuscript. The reviewer’s specific comments have been addressed as follows:

1.) Abstract can be improved by adding the one line about the "Cluster 1.2 genomes in all the three full-length LSDV."  

Answer: The reference to Cluster 1.2 has been included

2.) Was cattle sample(s) part of the current research? While reading the manuscript it is creating little confusion (Line 194-195). Please mention it in the materials and methods section.

Answer: All the samples used were submitted to the ARC-OVR for molecular laboratory confirmation of LSD. The primary focus of the study was the wildlife samples, but additional samples from cattle submitted in 2010 were included since the originally submitted tissue material were still in a good state to be used for PCR and Sanger sequencing.

3.) What are the chances of introducing DNA mutation while using Dreamtaq DNA polymerase as opposed to high fidelity DNA polymerase? Is it of any major concern in the present study?

Answer: No, it is not a concern for the Sanger sequencing performed in this study. The fidelity of the polymerase is sufficiently adequate so as not to introduce mutations during Sanger sequencing reactions. This could have been of concern had the amplicons been submitted to a NGS technology platform.

 4.) As current data is strongly suggestive of LSDV is being transmitted between cattle and wildlife. Was LSDV detection attempted in the cattle/wildlife/vectors?

Answer: Information on the number of samples tested is included in the Results section. All the samples used were submitted to the ARC-OVR for molecular laboratory confirmation of LSD. Since these are predominantly for disease confirmation, surveillance or export purposes, the samples are from cattle and wildlife, but not from insect vectors.

5.) A separate "conclusions" heading may help the readers. Conclusions may be limited to the current experimental findings.  

Answer: The discussion was changed to focus on the work and the results obtained.

Specific comments:

Line 4, 6, 7: full stops can be deleted.

Answer: Changes made as suggested

Line 123, 124: use "μL" instead of "ul". Use a single space between a number and the unit.

Answer: Changes made as suggested

Line 127: use a single space between a number and the unit.

Answer: Changes made as suggested

Line 138, 139: use 10 mM instead of 10mM, pH 8 instead of pH8, 10 mM instead of 10mM, 5 mM instead of 5mM. 

Answer: Changes made as suggested

Line 141, 142, 143, 144: Keep a space between a number and °C.

Answer: Changes made as suggested

Line 142: use 25 μL instead of 25μl.

Answer: Changes made as suggested

Line 145-148: use space between a number and unit μL. Use μL instead of μl.

Answer: Changes made as suggested

Line 145: use 10 mM instead of 10mM, pH 8.0 instead of pH8.0, 1 mM instead of 1mM. 

Answer: Changes made as suggested

Line 146: use 500 pg instead of 500pg.

Answer: Changes made as suggested

Line 147: use 4 °C instead of 4°C.

Answer: Changes made as suggested

Line 148: use 5 M instead of 5M. 

Answer: Changes made as suggested

Line 152: please define "vDNA."

Answer: The viral DNA (vDNA) was defined

Line 153: use 200 ng instead of 200ng, 300 bp instead of 300bp.

Answer: Changes made as suggested

Line 155: use 25 ng instead of 25ng.

Answer: Changes made as suggested

Line 174: "is presented in" can be deleted

Answer: Changes made as suggested

Table 1: Date column: Aug can be used to create uniformity instead of August.

Answer: All dates were changed to include the full name

Table 1: Animal species and sample type column: what was the sample type for raw LSDV_SB2366-FS_RSA_2011? Suggesting a colon between animal species sample type. Can "common animal name" be used as column heading instead of animal species?

Answer: A heading indicating that is the common name of the animal, has been included

Table 1: Full form of NW is missing in the table abbreviations.

Answer: This was included in the abbreviation list.

Following sections between text and references need to be added to provide more transparency and clarity for the audience. 

Author Contributions

Institutional Review Board Statement

Informed Consent Statement

Answer: All of the sections have been included.

Reviewer 3 Report

Comments and Suggestions for Authors

The ms "Complete sequences of lumpy skin disease virus genomes obtained directly from lesions on wildlife game species in South Africa" presents a molecular study conducted to detect and characterize LSDV in samples collected from wild species in South Africa (springbock, impala and giraffe). Although providing some useful updates on these poxviruses, the ms requires major revision before being published. Find below specific comments:

English language should be revised and can be improved throughout the ms. Try to not repeat concepts and sentences.

Title might be improved (for example: "Detection and genome sequencing of Lumpy skin disease virus from wildlife species in South Africa")

When referring to samples, try not to repeatedly use "submitted", for instance replace it with "collected" or other terms, and adapt language to the context.

M&M. Section 2.1 can be re-named "Sample collection and analysis"

Lines 132-133. How was the phylogenetic analysis conducted in MEGA? Why GTR model was used? How was the model selected? This should be clarified and explained in M&M.

Results. Section 3.1 should be shortened and simplified, some parts should be actually moved to M&M. Generally speaking, do not mix Materials and Methods with Results, do not repeat already mentioned methods in the Results, and do not repeat parts of the Introduction in the Discussion section.

Throughout the ms, it is not always clear which samples were actually analyzed in the present work, and historical reviews are mixed with the current analysis, try to be clear on this point.

Fig. 2. Map should be put into context (showing Africa or at least South Africa and then zooming to the sampling area).

Fig. 3. Bootstrap values obtained in the phylogenetic analysis should be shown at the nodes of tree branches.

Discussion section is confused and not clear, and language needs revision. Although interesting, try not to write a story (see comments above, about repeating Introduction in the Discussion), but a clear discussion of your results (as in the last paragraphs), inferences on LSDV presence and characterization, and implications for management and conservation of the monitored wildlife species.

Comments on the Quality of English Language

English language should be revised and can be improved throughout the ms.

Author Response

Thank you kindly for reviewing the manuscript and providing suggestions to improve the manuscript. The reviewer’s specific comments have been addressed as follows:

English language should be revised and can be improved throughout the ms. Try to not repeat concepts and sentences.

Answer: The manuscript has been subjected to extensive editing.

Title might be improved (for example: "Detection and genome sequencing of Lumpy skin disease virus from wildlife species in South Africa")

Answer: Title changed as suggested

When referring to samples, try not to repeatedly use "submitted", for instance replace it with "collected" or other terms, and adapt language to the context.

Answer: Changes have been made to the manuscript

M&M. Section 2.1 can be re-named "Sample collection and analysis"

Answer: Changes have been made to the manuscript

Lines 132-133. How was the phylogenetic analysis conducted in MEGA? Why GTR model was used? How was the model selected? This should be clarified and explained in M&M.

Answer: Previous publications where the model was tested and selected using a similar data set was indicated.

Results. Section 3.1 should be shortened and simplified, some parts should be actually moved to M&M. Generally speaking, do not mix Materials and Methods with Results, do not repeat already mentioned methods in the Results, and do not repeat parts of the Introduction in the Discussion section.

Answer: The section has been moved to the Materials and methods as suggested

Throughout the ms, it is not always clear which samples were actually analyzed in the present work, and historical reviews are mixed with the current analysis, try to be clear on this point.

Answer: Changes have been made to the manuscript

Fig. 2. Map should be put into context (showing Africa or at least South Africa and then zooming to the sampling area).

Answer: A complete map of South Africa is included in Figure 2, but this has been added to the figure legend.

Fig. 3. Bootstrap values obtained in the phylogenetic analysis should be shown at the nodes of tree branches.

Answer: New Figure 3 has been included indicating the Bootstrap values.

Discussion section is confused and not clear, and language needs revision. Although interesting, try not to write a story (see comments above, about repeating Introduction in the Discussion), but a clear discussion of your results (as in the last paragraphs), inferences on LSDV presence and characterization, and implications for management and conservation of the monitored wildlife species.

Answer: The discussion was changed to focus on the work and the results obtained.

Round 2

Reviewer 3 Report

Comments and Suggestions for Authors

The ms "Detection and genome sequencing of lumpy skin disease viruses from wildlife game species in South Africa" has definitely been improved following the first review, but it still needs revision.

The paper is still too long, not clear throughout and needs to be shortened.

It is better not to start Results with the Figures but with text results, with the Fig. embedded when cited in the text.

English language, although improved, still needs further revision throughout the ms, which is highly recommended.

Comments on the Quality of English Language

English language should be revised and can be improved throughout the ms.

Author Response

Thank you kindly for the suggestions to improve the manuscript.

The figures were move into the text as suggested and the manuscript was revised by an external language editor.
